# Approaching boiling point stability of an alcohol dehydrogenase through computationally-guided enzyme engineering

Friso S Aalbers[1,2], Maximilian JLJ Fürst[1,3], Stefano Rovida[2], Milos Trajkovic[1], J Rubén Gómez Castellanos[2], Sebastian Bartsch[4], Andreas Vogel[4], Andrea Mattevi[2], Marco W Fraaije[1]*

[1]Molecular Enzymology Group, University of Groningen, Groningen, Netherlands; [2]Department of Biology and Biotechnology "L. Spallanzani", University of Pavia, Pavia, Italy; [3]MRC Laboratory of Molecular Biology, Francis Crick Avenue, Cambridge Biomedical Campus, Cambridge, United Kingdom; [4]c-LEcta GmbH, Leipzig, Germany

**Abstract** Enzyme instability is an important limitation for the investigation and application of enzymes. Therefore, methods to rapidly and effectively improve enzyme stability are highly appealing. In this study we applied a computational method (FRESCO) to guide the engineering of an alcohol dehydrogenase. Of the 177 selected mutations, 25 mutations brought about a significant increase in apparent melting temperature ($\Delta T_m \geq$ +3 °C). By combining mutations, a 10-fold mutant was generated with a $T_m$ of 94 °C (+51 °C relative to wild type), almost reaching water's boiling point, and the highest increase with FRESCO to date. The 10-fold mutant's structure was elucidated, which enabled the identification of an activity-impairing mutation. After reverting this mutation, the enzyme showed no loss in activity compared to wild type, while displaying a $T_m$ of 88 °C (+45 °C relative to wild type). This work demonstrates the value of enzyme stabilization through computational library design.

*For correspondence:
m.w.fraaije@rug.nl

## Introduction

The oxidation of alcohols and reduction of ketones in nature is primarily governed by alcohol dehydrogenases (ADHs) (EC 1.1.1.1) (*Kavanagh et al., 2008*; *Oppermann et al., 2003*). There is a vast diversity of ADHs that can transform a large range of substrates, often with high regio- and enantio-selectivity. Most ADHs employ nicotinamide cofactors for catalysis. In particular, ketone reductions catalysed by ADHs are used in industry to synthesize chiral alcohols (*Nealon et al., 2015*; *Zheng et al., 2017*). Although ADHs have been mainly applied for ketone reductions, recent studies show promising results as biocatalysts for selective oxidations (*Solé et al., 2019*).

NAD(P)-dependent ADHs catalyse the transfer of a hydride from the reduced nicotinamide coenzyme to reduce ketones to alcohols, or the transfer of a hydride from alcohols to reduce NAD(P)$^+$. A large subset of alcohol dehydrogenases belongs to the family of short-chain reductases/dehydrogenases (SDRs) (*Kavanagh et al., 2008*). Enzymes from this family encompass around 250 amino acid-long sequences, typically form tetramers, use NAD(H) or NADP(H) as cofactor, and mainly act on secondary alcohols and ketones. The characteristic catalytic triad in SDRs is a combination of Ser-Tyr-Lys (*Bhatia et al., 2015*). These residues are located close to the C4 of the nicotinamide ring of the cofactor, at which the hydride is located or is transferred to. The catalytic triad, along with several residues in its vicinity, form a proton-relay system that either assists in the deprotonation of an

alcohol to initiate alcohol oxidation, or facilitates the protonation of a ketone substrate with concurrent hydride transfer, resulting in ketone reduction.

Most enzymes in nature have evolved to catalyse reactions within cells, in crowded, mild and aqueous conditions, and only for limited periods of time (*DePristo et al., 2005*; *Goldenzweig and Fleishman, 2018*). Moreover, enzymes that can be easily degraded after some time enable greater metabolic adaptability for the organism, for instance when a local resource depletes, or a signaling event occurs (*DePristo et al., 2005*; *Parsell and Sauer, 1989*). Therefore, many enzymes are only moderately stable. When considering enzymes for applications, this can lead to various difficulties in heterologous expression, degradation upon isolation, and poor biocatalyst performance. Enzymes that have the potential to be applied as biocatalyst are often insufficiently stable (*Bommarius and Paye, 2013*; *Woodley, 2019*). One such enzyme is ADHA, an alcohol dehydrogenase from *Candida magnoliae* DSMZ 70638. This enzyme was isolated for its ability to reduce several ketones with excellent enantioselectivities (*Table 1*). These catalytic features make ADHA attractive for industrial applications. Although heterologous expression of ADHA is good, the melting temperature of ADHA is moderate, with an apparent $T_m$ of 43 °C. If stability of ADHA could be improved, it would become a very attractive biocatalyst for selective oxidations or reductions.

Several studies over the past two decades have developed strategies to stabilize proteins and enzymes. A subset of these strategies specifically targeted the charge of the protein or enzyme surface or interface (*Bjørk et al., 2004*; *Gribenko et al., 2009*). More recently, the field has mostly aimed at the development of computational methods to predict mutations that improve the stability of proteins (*Malakauskas and Mayo, 1998*; *Murphy et al., 2012*; *Shah et al., 2007*) and/or enzymes (*Bednar et al., 2015*; *Borgo and Havranek, 2012*; *Korkegian et al., 2005*; *Moore et al., 2017*). Considering the large improvements in stability that were found using these methods with minimal screening work in the lab, such computational approaches are highly appealing.

Our goal was to improve the stability of ADHA through enzyme engineering, guided by a computational method. The requirement for this endeavour was to obtain a crystal structure of the enzyme. The structure is needed as input for our in-house developed method: 'framework for rapid enzyme stabilization by computational libraries' (FRESCO) (*Wijma et al., 2018*). This computational protocol incorporates two algorithms to calculate folding free energy ($\Delta G^{Fold}$, expressed in kJ mol$^{-1}$), and compares these calculations ($\Delta\Delta G^{Fold}$) between the wild-type structure and variants with a single mutation. The calculations are performed for all possible single mutations. The variants with the best scores are simulated with molecular dynamics, to inspect the flexibility and conformational stability. In the last round of enzyme engineering, mutations that improve the stability are combined to obtain a final robust variant. The main motivation to apply FRESCO is that it is relatively fast, because it greatly reduces the required screening effort compared to other enzyme engineering approaches.

Here, we present the results of the application of FRESCO on ADHA. This has led to a variant which has been mutated at 10 positions and displays a drastically increased thermostability ($T_m$ = 94.5 °C, $\Delta T_m$ = +51 °C). However, it was found that this engineered ADHA displayed very little activity. Elucidation of its crystal structure disclosed the structural basis for the inactivation: one particular mutation, in combination with other introduced mutations, resulted in a reorientation of a loop region through which binding of the nicotinamide cofactor is hampered. By reverting this mutation, a highly stable ($T_m$ = 88 °C) and active ADHA variant was obtained, demonstrating the power of using computationally predicted mutations in combination with structural analysis to engineer robust enzymes.

**Table 1.** Initial measurements of ADHA WT activity and selectivity with several ketones.

| Substrate (concentration) | $k_{obs}$[*] (U/mg) | Enantiomeric excess product (*ee*) |
|---|---|---|
| Ethyl acetoacetate (100 mM) | 4.1 | n.d. |
| Cyclohexanone (50 mM) | 3.2 | n.a. |
| Ethyl 4-chloro-3-oxobutanoate (COBA) (50 mM) | 0.3 | > 99% (*R*) |
| 4-Chloroacetophenone (4-CAP) (50 mM) | 2.4 | 97.2% (*S*) |

[*]$k_{obs}$ values are averages based on 2–3 replicates and for each average the error was smaller than 5%.

## Results

### Characterization of ADHA

Characterization of ADHA immediately indicated a high reductive activity towards a broad panel of ketones (*Table 1*) with high enantioselectivity (> 97% *ee*). This result illustrated that the enzyme could be applied for the synthesis of various chiral alcohols. Therefore, we concluded that ADHA has great potential for the application as a biocatalyst. However, we soon realized that most of the enzyme activity is lost in reactions of 8 h or longer, even at moderate temperatures such as 25 °C or 30 °C (data not shown). Therefore, we decided to employ FRESCO to generate a more robust variant.

### Crystal structure of wild-type ADHA

A prerequisite for FRESCO is the availability of a crystal structure of the target protein. Therefore, already in the early stages of the project, the untagged ADHA was tested for crystallization trials. The purified enzyme crystallized in many conditions and the enzyme three-dimensional structure was solved in the apo and NADP$^+$-bound state at 2.0 Å and 1.6 Å resolution, respectively (*Figure 1*, *Supplementary file 1A*). ADHA is a homotetramer as typically observed in most SDRs (*Kavanagh et al., 2008*). A search for homologous structures in the Protein Data Bank showed that the closest structural homologue is another alcohol dehydrogenase from *Candida magnoliae* (PDB entry 5MLN [*Tavanti et al., 2017*]; root-mean-square deviation of 1.0 Å for 238 Cα atoms with 64% sequence identity; *Figure 1*). The structure of ADHA bound to NADP$^+$ unveils, apart from a disordered nicotinamide moiety, a well-defined catalytic pocket with a size suited for small cyclic compounds such as cyclohexanol, a good substrate of the enzyme.

### FRESCO computational predictions

The obtained crystal structure of the wild-type tetrameric ADHA allowed us to predict stabilizing mutations through a structure-based computational approach using FRESCO (*Wijma et al., 2018*). A detailed description of the procedure is available as a step-by-step protocol suitable for biochemists with minimal computational experience and all the scripts are deposited online (*Wijma et al., 2018*). Licenses for the ΔG$^{Fold}$-determining software components (Rosetta and FoldX) are free for academic purposes, whereas our preferred software for modelling and MD simulations, YASARA, needs to be

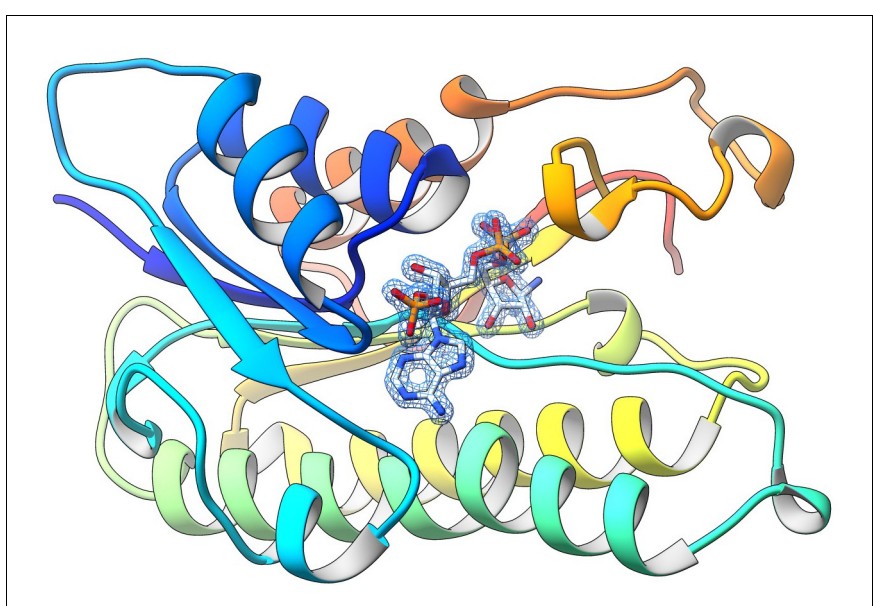

**Figure 1.** The crystal structure of ADHA. The figure highlights the final weighted 2Fo-Fc map for NADP$^+$ bound to a subunit of the wild-type ADHA (subunit A, contour level 1.2 σ). The nicotinamide moiety of the cofactor is disordered and was not included in the final model.

purchased. Depending on the size of the protein, a computer cluster may be necessary for the calculation-intensive parts of the predictions. After FRESCO's in silico screening, a final selection of mutants is expressed, purified and tested for stability in vitro. The experimental part has also been described in great detail (*Fürst et al., 2018*) and requires readily purchasable consumables and equipment commonly available in most labs.

Because the energy prediction algorithms Rosetta and FoldX do not accept non-proteinogenic residues and because the ligand-free enzyme represents the physiologically more relevant form, we used the crystal structure of ADHA without NADP$^+$ for the stabilizing mutation predictions. While FRESCO usually excludes residues of ligand-binding sites, we decided to include such residues for predicting stabilizing mutations. Because the enzyme is relatively small, we could still easily discard activity-impairing mutations in the experimental phase. We did not include the residues that form the canonical catalytic triad (T148, Y161, and K165), which are highly conserved residues and essential for enzyme activity. Curiously, unlike ADHA, the majority of SDRs have a serine at the position of T148. In fact, the catalytic triad of SDRs is usually designated as SYK (serine, tyrosine, lysine). Clearly, a catalytic threonine can also support redox catalysis in an SDR-type alcohol dehydrogenase, and such an alternative catalytic triad has also been observed before (*King et al., 2007*). The calculations for the 4503 possible mutations were ranked and mutations with a $\Delta\Delta G^{\text{Fold}} \leq -5$ kJ mol$^{-1}$ were briefly simulated with molecular dynamics (MD) simulations to probe the flexibility of the variant. The resulting 478 computer-generated mutant structures were visually inspected to remove chemically undesirable mutations (clashes, exposed hydrophobic residues, loss of hydrogen bonds), and highly flexible mutants (based on the short MDs) (*Wijma et al., 2018*). In addition, several positions (> 20) had more than four suggested beneficial mutations. In those cases, we chose two or three distinct suggested mutations. Finally, we selected 177 single mutations for experimental screening.

## In vitro analysis of mutant library

The mutant library was prepared using the QuikChange method following the published protocol (*Fürst et al., 2018*). After growth of the cells harbouring the genes with each mutation and inducing protein expression, the cells were lysed and the His-tagged ADHA variants were purified by affinity chromatography. The purified samples were desalted and the apparent $T_m$ was measured in an RT-PCR machine in duplicate. 26 mutants repeatedly did not give a clear signal, which may have been caused by insufficient expression. Of the 151 mutants for which a melting temperature could be measured, 52 had a similar $T_m$ when compared to wild-type ADHA ($-1 \leq \Delta T_m \leq +1$ °C), 43 were destabilizing ($\Delta T_m < -1$ °C), and 56 were stabilizing ($\Delta T_m > +1$ °C) (*Figure 2*). Of the 56 stabilizing mutants, 25 had a $\Delta T_m \geq +3$ °C. However, when measuring activity, 11 of these 25 mutants displayed very poor or no activity (*Supplementary file 1B*). For the next phase of the engineering project, we focused on 10 mutations that retained most of the activity and resulted in a significantly higher $T_m$ (labelled red in *Figure 2*).

## Combining mutations

The most stabilizing mutations were ranked according to their thermostability and combined by consecutively introducing them one by one (*Table 2*). The combined mutants (named MX, with X = the number of mutations) were made using the plasmid of the previous mutant (e.g. M4 was made from the plasmid of M3). Only in the case of the V193L and T194V mutations, the mutations were introduced in one step as these residues were next to each other (going from M7 to M8). Interestingly, the combined mutants showed superior expression compared to wild-type ADHA, and could be expressed at higher temperatures for shorter periods of time, such as 37 °C overnight (*Supplementary file 2A*). Beyond M4, the enzyme gave two melting points in the ThermoFluor measurements (indicated by the values in parentheses, *Table 2*), of which the first curve appears 10–11 °C before the second. The first curve is typically significantly lower in intensity compared with the second (major) curve (*Figure 2—figure supplement 1*) and may reflect a mild unfolding event such as the dissociation of the tetrameric enzyme into (active) dimers.

Gratifyingly, we observed that the mutations were additive, as they incrementally increased the enzyme thermostability. The additive effect is not perfect, as the theoretical maximal additive effect would be 56.5 °C, while M9 displays an improvement of 51.5 °C. This may indicate that some of the mutations are part of the same structural "soft spot" that triggers early unfolding events in the wild

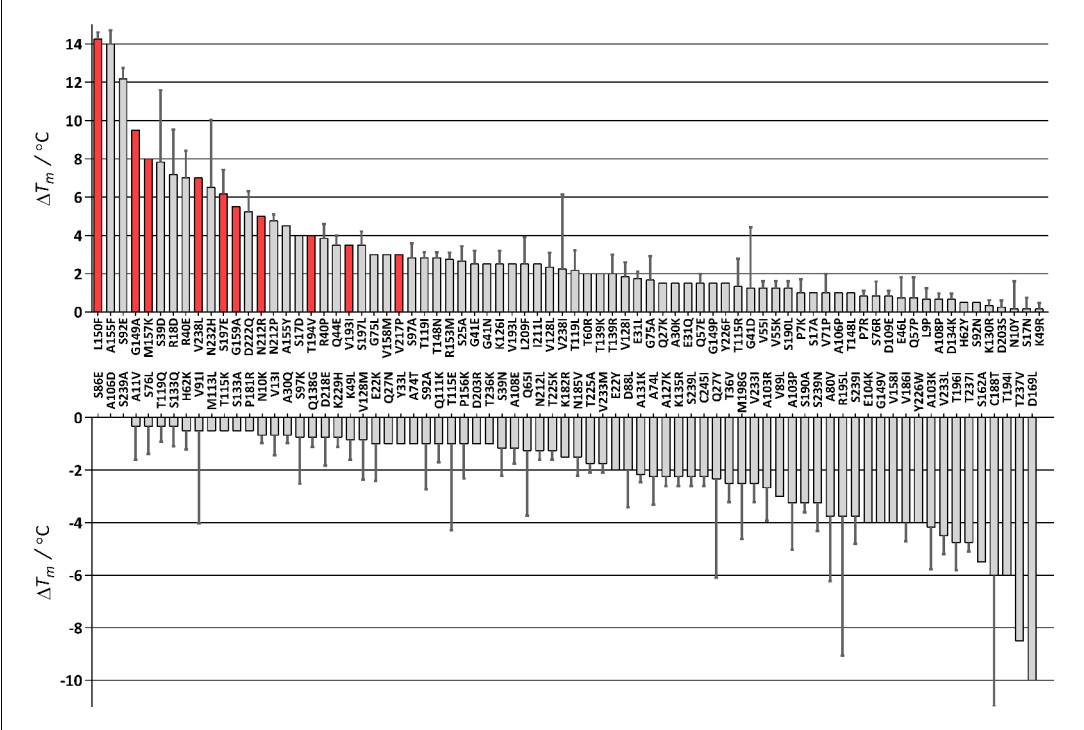

**Figure 2.** Difference in $T_m$ for 151 FRESCO-predicted ADHA mutants. The average of two measurements is given and the standard error. The $T_m$ of wild-type ADHA is 43 °C (set as 0). The 10 stabilizing mutations with a red bar were combined. Melting curves of wild type and the final mutant (M9*) are depicted in *Figure 2—figure supplement 1*.

The online version of this article includes the following figure supplement(s) for figure 2:

**Figure supplement 1.** Melting curves.

type. Still, an increase of > 50 °C by introducing 10 mutations was highly gratifying. Adding more hits (S25A, G41N, G75L) from the library beyond the 10$^{th}$ mutation did not increase the apparent melting temperature any further. This prompted us to investigate the M9 mutant in more detail. We found that the M9 enzyme retained catalytic activities, though with a 100-fold higher $K_M$ for NADP$^+$ (1.0 mM, compared to 9.5 µM) and a 4-fold decreased $k_{cat}$ (0.25 s$^{-1}$, compared to 1.1 s$^{-1}$) (*Figure 3*). We suspected that this would be the result of a shift in optimal temperature for activity, due to the presumed higher rigidity of this highly stabilized mutant. But even at higher temperatures ($\geq$ 40 °C), the activity of the M9 mutant did not drastically change. Clearly, the multiple mutations did result in a highly stable but catalytically impaired enzyme and it was essential to elucidate the specific structural cause of this.

## Crystal structure of the M9 mutant

To investigate the structural alterations caused by the stabilizing mutations and their effect on enzyme catalysis, we decided to crystallize mutant M9. As with the wild-type ADHA, several conditions gave crystals of the M9 enzyme, enabling determination of its structure at 2.6 Å resolution. Although the crystallization conditions contained 1.0 mM NADP$^+$, there was no electron density for any ligand in the M9 structure. This was not surprising, considering the high $K_M$ (1.0 mM) for NADPH of this mutant. When inspecting the M9 structure, it is worth noting that all the introduced mutations are in the same area of the enzyme three-dimensional structure. Specifically, the mutations are located centrally on each of the two flat faces of the homo-tetramer, where two opposing subunits come together (*Figure 4A*). Most mutations are located close to the tetramer centre (*Figure 4B*) and many are surface-exposed, or surface-near. Superposition of the wild-type ADHA and M9 tetramers revealed a very close structural similarity with a root-mean-square deviation of 0.58 Å for 956 Cα atoms. The only considerable backbone shift (up to 3.1 Å) concerns loop 196–214 (*Figure 4B,C*).

**Table 2.** ADHA mutants with highest $\Delta T_m$ and retained activity.

In parentheses the temperature of the first unfolding event (minor peak) is given. Michaelis-Menten kinetics for wild type and M9 are shown in *Figure 3*.

| Single mutants | $k_{obs}$* (U/mg) | $T_{m\ app}$ (°C) | $\Delta T_m$ (°C) |
|---|---|---|---|
| Wild type | 0.6 | 43.0 | - |
| L150F | 0.5 | 57.25 ± 0.25 | 14.0 ± 0.25 |
| G149A | 0.4 | 52.5 ± 0 | 9.5 ± 0 |
| M157K | 0.4 | 51.0 ± 0 | 8.0 ± 0 |
| S197E | 0.5 | 49 ± 1 | 7.5 ± 1 |
| V238L | 0.4 | 50.0 ± 0 | 7.0 ± 0 |
| G159A | 1.2 | 48.5 ± 0 | 5.5 ± 0 |
| N212R | 1.1 | 48.0 ± 0 | 5.0 ± 0 |
| T194V | n.d. | 47.0 ± 0 | 4.0 ± 0 |
| V193L | n.d. | 46.5 ± 0 | 3.5 ± 0 |
| V217P | n.d. | 46.0 ± 0 | 3.0 ± 0 |
| **Combination mutants** | $k_{obs}$* (U/mg) | $T_{m\ app}$ (°C) | $\Delta T_m$ (°C) |
| M2 (L150F + M157K) | 0.4 | 64.0 ± 0 | 21.0 ± 0 |
| M3 (M2 + S197E) | 0.1 | 69.0 ± 0 | 26.0 ± 0 |
| M4 (M3 + V238L) | 0.2 | 72 (62) ± 0.25 | 29.0 ± 0.25 |
| M5 (M4 + N212R) | 0.2 | 75.5 (64) ± 0 | 32.5 ± 0 |
| M6 (M5 + G149A) | 0.1 | 81.0 (69) ± 0 | 38.0 ± 0 |
| M7 (M6 + G159A) | 0.1 | 85.0 (74) ± 0 | 42.0 ± 0 |
| M8 (M7 + V193L + T194V) | 0.05 | 90.0 (81) ± 0.25 | 47.0 ± 0.25 |
| M9 (M8 + V217P) | 0.03 | 94.5 (84) ± 0 | 51.5 ± 0 |
| M9* (M9 - S197E) | 0.8 | 88.0 (78.5) ± 0 | 45.0 ± 0 |

*$k_{obs}$ values are averages based on 2–3 replicates and for each average the error was smaller than 5% (between ± 0.0015–0.04 U/mg). Cyclohexanol was used as substrate.

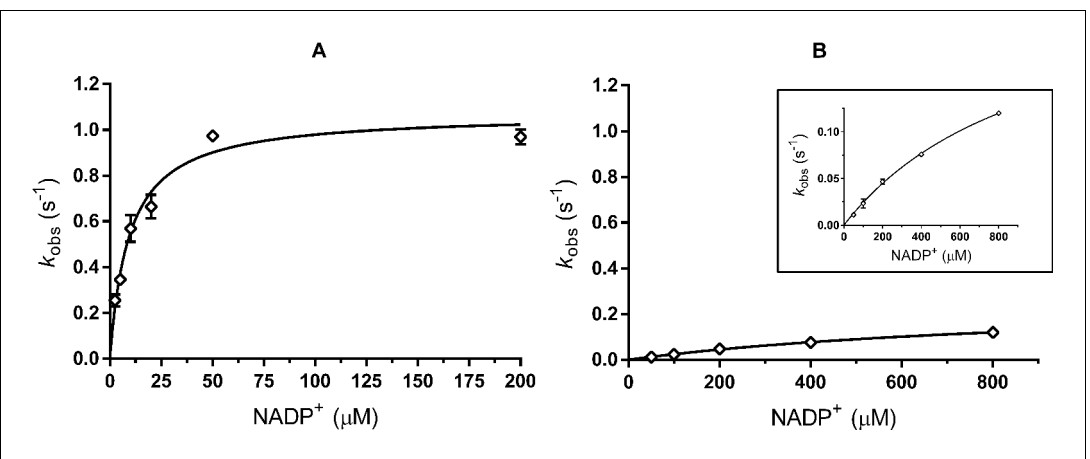

**Figure 3.** Michaelis-Menten plots for kinetics with NADP$^+$. (**A**) ADHA wild type (**B**) M9 mutant. Note that the X-axis scaling is different. The inset of B presents the same data with a different Y-axis scaling. Plots are fitted with Michaelis-Menten in GraphPad prism 6.07.

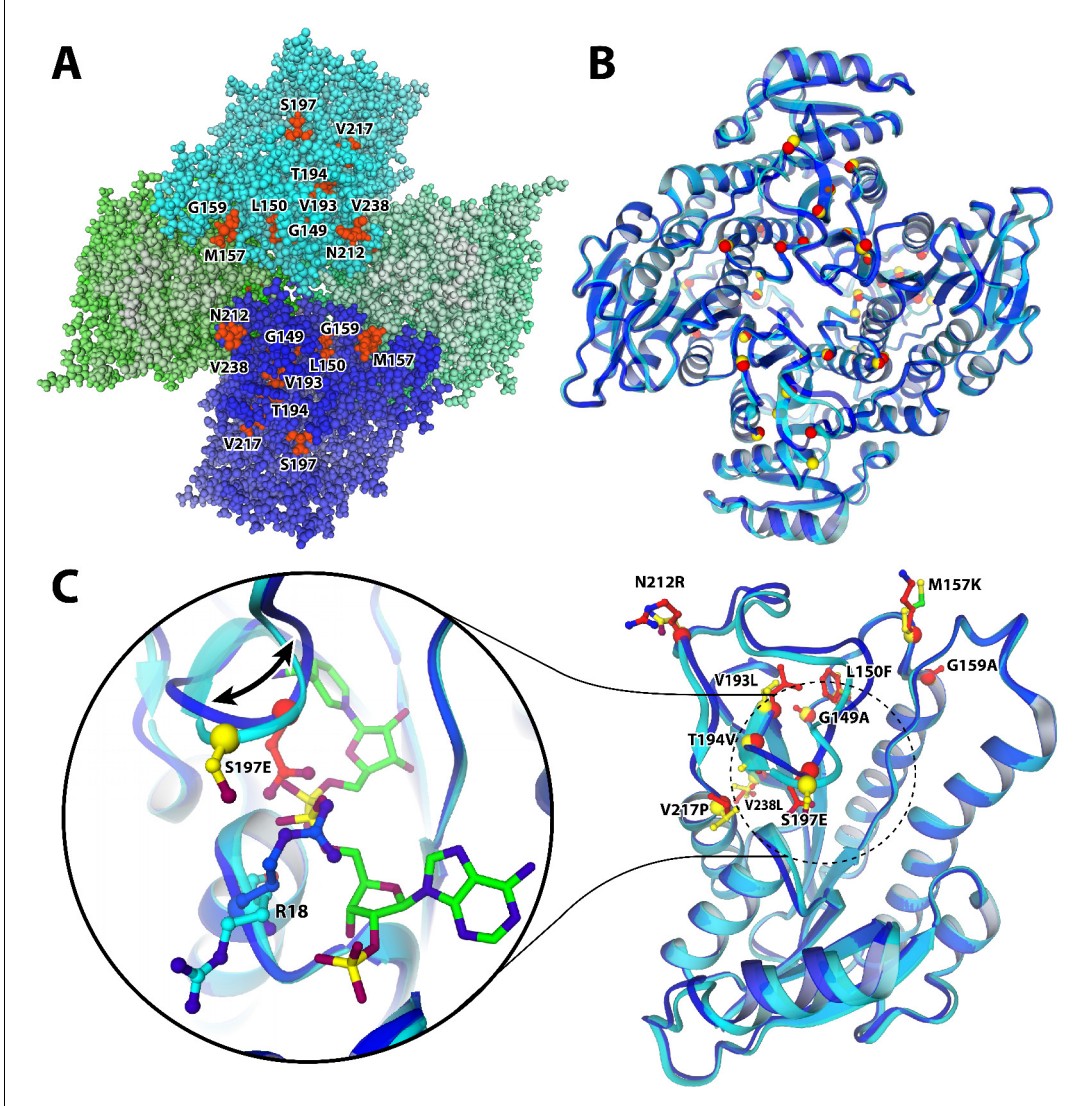

**Figure 4.** Structure of the M9 mutant of ADHA with mutated resides highlighted. (**A and B**) quaternary structure of M9. The tetramer is organized such that the N-termini are on the outside (on the edge of the top-down view of A and B), whereas the C-termini all point inwards; which is where most and the most stabilizing mutations were found. (**A**) M9 structure with all atoms represented as balls. The four monomers are shaded in various colours, highlighting the particular clustering of the observed stabilizing mutations. (**B**) The structure as ribbon model, superimposing the mutant (blue ribbon, red spheres indicate mutated residue) and the wild type (cyan ribbon, yellow spheres). (**C**) Colour scheme as in B. The loop (196-214) that is dislocated as a result of the S197E mutation, compared to the structure of wild-type ADHA. The shift is accompanied by a flip of R18 into the NADP-binding pocket, likely due to an electrostatic attraction from the mutant glutamate. As a result, the cofactor (green carbons) is only bound in the wild type, while absent from the mutant structure.

Residues 196–198 in this loop are involved in binding the NADP$^+$ pyrophosphate group as shown by the crystal structure of the wild-type ADHA bound to NADP$^+$. The observed structural alteration in this particular loop immediately provides an explanation for the increase of $K_M$ for this cofactor in M9: the introduced E197 seems to occupy the NADP$^+$ pyrophosphate binding site. Along with the shift of the loop, the carboxylate moiety of the S197E mutation induces an interaction with the guanidinium of R18 (*Figure 4C*) – a residue that crucially binds the 2' phosphate of NADP$^+$ in the wild-type structure. While the favourable electrostatics of this new interaction is a likely explanation for the mutation's stability effect, it also explains the lower affinity for NADP$^+$ (and the absence of the cofactor in the mutant's crystals), as the salt bridge neutralizes R18 and occupies the NADP$^+$ binding pocket.

Closely located to the S197E mutation are several other introduced mutations: G149A, L150F, V193L, T194V, and V217P. These mutations cause rather small alterations by exchanging a hydroxyl for a methyl group or slightly increasing the bulkiness of aliphatic side chains. These residues are part of a cluster of hydrophobic residues adjacent to the loop 196–214 and involve several aliphatic (I20, P191, M198, I215, I220) and aromatic (F207, W214, Y244) side chains. G149A, L150F (the most stabilizing mutation of all), V193L, T194V, and V217P all contributed substantially to thermostability, with up to 14 °C increase in the $T_m$ values at the single mutant level (*Table 2*). The structural analysis clearly shows that improved side-chain packing and hydrophobic interactions are the likely source of their stabilizing effect. Another mutation causing improved hydrophobic packing is G159A ($T_m$ increase of 5.5 °C) that locates close to I102, F171, and L175. V238L (+7 °C) is a unique case, promoting more extended interaction of the subunits within the tetramer (*Figure 5*). Curiously, the mutation does so by allowing a hydrophobic interaction with itself at the dimer interface. V217P is a seemingly conservative mutation at the beginning of an α-helix and yet caused a significant effect (+3 °C; *Table 2* and *Figure 4C*). Prolines often occur at 'helical capping' positions by restraining the backbone conformation and positioning a substituted backbone amino group that is unable to form H-bonding interactions (*Richardson and Richardson, 1988*). The V217P mutation follows this trend. Among the identified mutations, M157K is the only one that replaces a hydrophobic residue with a hydrophilic and charged side chain (*Figure 4C*). It has a large (+8 °C; *Table 2*) effect on protein stability, which very likely arises from the removal of a solvent-exposed hydrophobic group and its replacement with the charged and hydrophilic amine of a lysine residue. Lastly, N212R is possibly the most puzzling mutation. It positions an arginine side chain in a fully solvent-exposed location (*Figure 4C*) and its conspicuous (+5 °C) stability can be tentatively attributed to the more pronounced polarity of an arginine side chain as compared to an asparagine (*Szilágyi and Závodszky, 2000*).

In summary, the detailed structural analysis of the mutation sites reveals a consistent pattern concerning the stabilizing effects: all stabilizing mutations of M9 promote hydrophobic packing of side chains or install highly polar and charged groups on fully solvent-exposed positions on the protein surface. The crystal structure of M9 also provided a clue to explain the rather low activity of M9.

## Rescue of activity

Since we observed and suspected that the S197E mutation recruits R18 into the cofactor binding pocket and thereby impairs NADP$^+$ binding (*Figure 4C*), we reverted this mutation. The 9-fold M9* mutant (M9 without S197E) was purified and characterized. As expected, we observed a decrease of 6.5 °C in $T_m$ for M9* compared to M9 (*Table 2*). Yet, it was satisfying that M9* displayed proper enzyme activity: the $K_M$ for NADP$^+$ of M9* has the same value as wild type (9.6 µM), and its $k_{cat}$ at 25 °C was only slightly lower (*Table 3*, *Figure 6*). With this final stabilized and active variant at hand, some more characterization experiments were carried out to probe its biocatalytic properties.

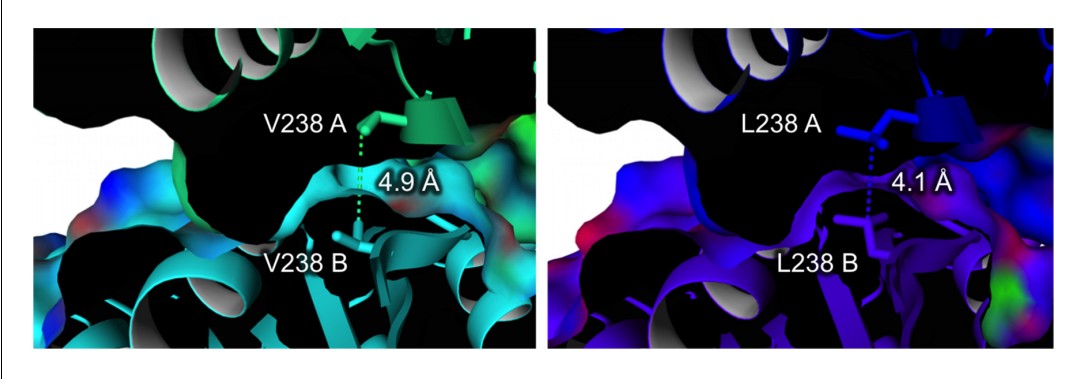

**Figure 5.** Dimer interface with the V238L mutation ($\Delta T_m$ = 7 °C). (**A and B**) indicate the different monomers in the ADHA tetramer.

**Table 3.** Characteristics of wild-type, M9, and M9* ADHA.

Activity measurements were performed at 25 °C in duplicate or triplicate and the respective Michaelis-Menten plots are depicted in *Figure 6*. Melting curves are depicted in *Figure 2—figure supplement 1*. Conversions were performed with 5 µM of ADHA and 10 mM of prochiral ketone substrate: ethyl 4-chloro-3-oxobutanoate (COBA) and 4-chloroacetophenone (4-CAP). Details and chromatograms are provided in *Supplementary file 4*.

| Enzyme | $T_m$ (°C) | $k_{cat}$ ($s^{-1}$) | $K_{M,NADP^+}$ (µM) | $k_{cat}/K_{M,NADP^+}$ ($s^{-1}$ $mM^{-1}$) | Conversion and *ee* (COBA) | Conversion and *ee* (4-CAP) |
|---|---|---|---|---|---|---|
| ADHA WT | 43.0 ± 0 | 1.1 ± 0.04 | 9.5 ± 1.2 | 116 ± 35 | > 99% <br> > 99% *ee* (R) | > 99% <br> 97.2% *ee* (S) |
| M9 | 94.5 ± 0 | 0.27 ± 0.02 | 1040 ± 108 | 0.26 ± 0.2 | n.d. | n.d. |
| M9* | 88.0 ± 0 | 0.7 ± 0.01 | 9.6 ± 0.7 | 73 ± 14.5 | > 99% <br> > 99% *ee* (R) | > 99% <br> 98% *ee* (S) |

By measuring the alcohol oxidation activity at various temperatures, a largely shifted and wider temperature optimum was found for M9* (*Figure 7A*). The enzyme shows highest activity at 55–60 °C. Only at temperatures > 75 °C, its activity decreases significantly. Still, the enzyme is also active at 85 °C, whereas wild-type ADHA displays hardly any activity > 40 °C. These data suggest that only the major unfolding event (at a relatively high temperature) observed by ThermoFluor results in inactivation of M9*. The apparent melting temperature was also measured in the presence of 0.5 mM NADPH. The M9 mutant showed a +3 °C increase, which is much smaller than the drastic +9 °C improvement in the $T_m$ of the wild type upon NADP$^+$ binding. We suspect that the regions that are stabilized through binding NADPH in the wild-type ADHA are partially stabilized by the FRESCO mutations, hence the relatively small gain in stability with NADPH as ligand for M9*.

To evaluate the stability over time, wild type and M9* ADHA were incubated at 37 °C, and their oxidation activity was measured at several time points (*Figure 7B*). As wild-type ADHA has a $T_m$ at 43 °C, it was losing activity steadily over the first couple of hours, with a half-life of around 2 hr. After 22 h at 37 °C, the remaining activity of wild-type ADHA was only 8.7%,

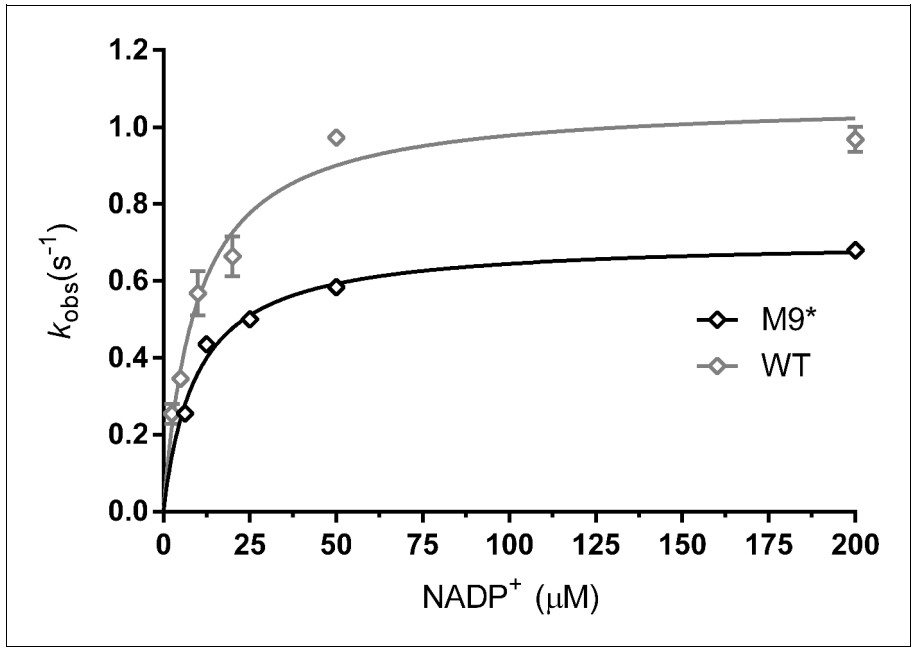

**Figure 6.** Michaelis-Menten plots for kinetics with NADP$^+$. ADHA wild type (grey, triangles) and M9* mutant (M9 with S197E reverted) (black, diamonds). Plots are fitted with the Michaelis-Menten equation in GraphPad prism 6.07.

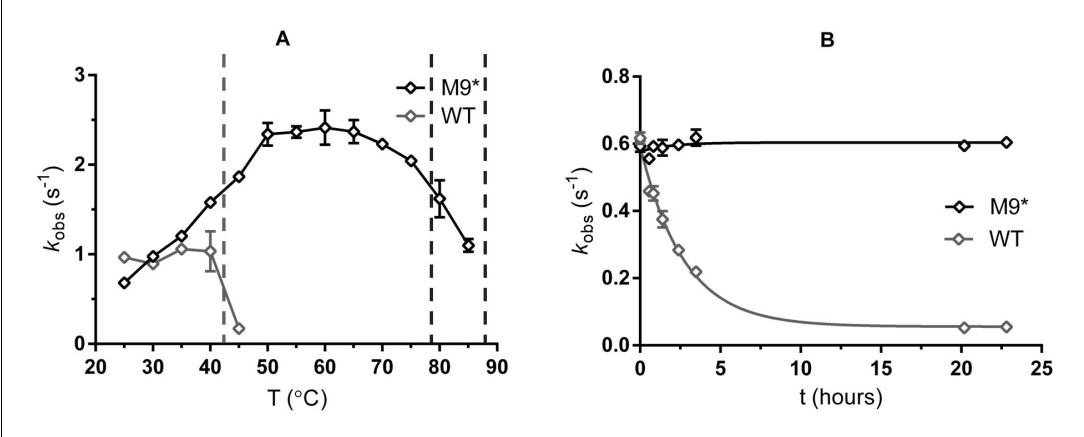

**Figure 7.** Properties of wild-type and M9* ADHA. (**A**) Temperature-activity profile using cyclohexanol as substrate. The dashed lines indicate the $T_m$ of the wild type (at 43 °C) and apparent melting temperatures of M9* (78.5 °C and 88 °C) (**B**) Enzyme activity monitored over time at 37 °C (buffer composition: 50 mM Tris-HCl pH 7.5). *Figure 7—figure supplement 1* depicts the enzyme activity over time for 18 days.

The online version of this article includes the following figure supplement(s) for figure 7:

**Figure supplement 1.** Long-term stability.

whereas the M9* still displayed full activity. Only after 11 days at 37 °C, the M9* ADHA lost 50% of its activity (*Figure 7—figure supplement 1*). The stability of the mutant was largely retained also in the presence of co-solvents, with isopropanol showing the largest $\Delta T_m$ (−21 °C; *Table 4*). This tolerance to the presence of 20% cosolvent is quite useful, as it enables higher concentrations of hydrophobic substrates.

## Discussion

With the aim of creating a robust ADHA variant, we applied FRESCO and reached a 9-fold mutant that shows a significantly improved thermostability with largely retained activity (*Table 3*). This study achieved the highest improvement of melting temperature of all FRESCO studies to date. Up to now, there have been six studies that applied FRESCO, and it has shown effective improvements in stability with $T_m$ differences up to +35 °C (*Table 5*). In particular, enzymes with higher order quaternary structures (e.g. dimeric and tetrameric enzymes) gained significant improvements of +28 °C and +35 °C (*Table 5*). Dimers and tetramers may gain a lot of stability from having subunit-subunit interfaces that stabilize hydrophobic regions of the enzyme. In contrast, enzymes containing cofactors (oxidases, Baeyer-Villiger monooxygenases) showed relatively moderate improvement: the FAD-containing HMF oxidase and cyclohexanone monooxygenase were improved by only 12–13 °C (*Table 5*). This may due to the fact that the employed computational algorithms (Rosettaddg and FoldX) cannot take into account the flavin-protein interactions (*Wijma et al., 2018*).

The initial FRESCO calculations and predictions for ADHA had a good hit rate: of the tested 177 mutations, 39 had a stabilizing effect of > 2 °C (22%). This is in line with previous uses of FRESCO, in

**Table 4.** Final melting temperatures ($T_m$) of M9* in various cosolvents (20% v/v).

| Cosolvent | $T_m$ (°C) | $\Delta T_m$ (°C) |
|---|---|---|
| - | 88.0 ± 0 | - |
| Methanol | 76.0 ± 0.5 | −12 |
| Ethanol | 71.5 ± 0.5 | −16.5 |
| Isopropanol | 67.0 ± 0.5 | −21 |

**Table 5.** Studies that have applied FRESCO for stabilization of enzymes, to date.

| Enzyme | Abbreviation | Size (aa) | Quaternary structure | $\Delta T_m$ | Reference |
|---|---|---|---|---|---|
| Limonene epoxide hydrolase | LEH | 149 | Dimer | +35 ˚C | (*Wijma et al., 2014*) |
| Haloalkane dehalogenase | LinB | 250 | Monomer | +22 ˚C | (*Floor et al., 2014*) |
| Hydroxymethyl furfural oxidase | HMFO | 525 | Monomer | +12 ˚C | (*Martin et al., 2018*) |
| Peptide amidase | PAM | 508 | Monomer | +23 ˚C | (*Wu et al., 2016*) |
| Halohydrin dehalogenase | HheC | 254 | Tetramer | +28 ˚C | (*Arabnejad et al., 2017*) |
| Cyclohexanone monooxygenase | CHMO | 529 | Monomer | +13 ˚C | (*Fürst et al., 2019*) |
| Glucose oxidase | GOX | 605 | Dimer | +8.5 ˚C | (*Mu et al., 2019*) |
| ω-Transaminase | ω-TA | 455 | Dimer | +23 ˚C | (*Meng et al., 2020*) |
| Short-chain dehydrogenase | ADHA | 246 | Tetramer | +45 ˚C | This work |

which 20–40% of the FRESCO-predicted mutations were found to be stabilizing (*Fürst et al., 2018*; *Wijma et al., 2018*). To see if the calculated energy values correlate with the observed $\Delta T_m$, the data were plotted and statistically analysed (*Supplementary file 3*). The Rosetta calculations bore no relation to the measured melting temperatures, whereas for the FoldX values some correlation was found (Pearson r = 0.3, p < 0.02). This somewhat better predictive quality of FoldX as compared to Rosetta has been described before (*Buß et al., 2018*). Still, 4/9 of the selected mutations were not found by FoldX, despite being highly stabilizing (*Supplementary file 3A*). Clearly, the combination of both calculations, together with the MD simulations in the FRESCO workflow, results in a satisfactory rate of predicted stabilizing mutations.

It could be argued that for the computational predictions the residues in close contact with the NADP$^+$ should be excluded, to avoid the complications that were found with the S197E mutation. During the visual screening of the single mutations with MD simulations the introduced glutamic acid was pointing outwards into the solvent, giving the impression of a somewhat harmless, surface-exposed mutation. The interaction it would have with R18, when in the context of the other introduced mutations, was impossible to predict. This newly introduced interaction was blocking binding of the nicotinamide cofactor. Yet, the 10-fold mutant (M9) is still active and could be of use when the enzyme needs to be used under challenging conditions. It also shows the power of combining FRESCO with a thorough structural inspection of the generated thermostable mutant. By structural analysis, the effects of individual mutations can be evaluated and, when needed, alternative structure-inspired mutations can be prepared. In this study, it led to the 9-fold M9* mutant of ADHA, which is highly thermostable, solvent tolerant, and active.

This study demonstrates how a computational method can give guidance for a path from a moderately stable enzyme to a highly robust enzyme, suitable for industrial settings with long reaction times and cosolvents. Although FRESCO requires some screening effort, the method greatly reduces the time and screening amount compared to directed evolution or semi-rational approaches. Moreover, since individual stabilizing mutations are combined, there is more certainty and freedom compared to computational methods that suggest stable variants with multiple mutations (*Goldenzweig et al., 2016*; *Musil et al., 2017*).

## Conclusions

A highly robust ADH was engineered through computational library design. It is the highest improvement in stability obtained with this method so far, with an improvement of $\Delta T_m$ +45 ˚C. Although some mutations might be avoided in the future, through careful consideration of cofactor binding, the nine selected mutations caused no significant decrease in activity or NADP$^+$ binding. This method can be effective to stabilize enzymes for industrial applications, in particular if the enzyme has a higher order quaternary structure.

# Materials and methods

## Key resources table

| Reagent type (species) or resource | Designation | Source or reference | Identifiers | Additional information |
|---|---|---|---|---|
| Strain, strain background (*Escherichia coli*) | NEB 10-beta chemically competent *E. coli* | New England Biolabs https://www.neb.com/ | C3019I | |
| Sequence based reagent | pBAD Golden gate vector (N-terminal 6xHis, araC, bla) | This study | | Molecular Enzymology Group, University of Groningen |
| Polymerase | PfuUltra II Hotstart PCR Master Mix | Agilent Technologies https://www.agilent.com/ | 600852 | |
| Commercial assay or kit | Ni chromatography resin | GE Healthcare Life Sciences https://www.gelifesciences.com/ | | |
| Software, algorithm | Rosetta | Rosetta Commons https://www.rosettacommons.org/ | RRID:SCR_015701 | |
| Software, algorithm | FoldX | FoldX http://foldxsuite.crg.eu/ | RRID:SCR_008522 | |
| Software, algorithm | YASARA | YASARA Biosciences GmbH http://www.yasara.org/ | RRID:SCR_017591 | |
| Software, algorithm | FRESCO scripts | https://groups.google.com/forum/#!forum/fresco-stabilization-of-proteins | | |
| Software, algorithm | GraphPad Prism | GraphPad Prism https://graphpad.com | RRID:SCR_015807 | Version 6 |

## Materials, strains and chemicals

Oligonucleotide primers for cloning and mutagenesis were ordered from Sigma-Aldrich. For amplification of the gene inserts or to perform QuikChange mutations, PfuUltra II HotStart PCR (Polymerase) master mix was used, purchased from Agilent Technologies. Other chemicals were purchased from Sigma-Aldrich and Acros Organics. Precast native PAGE gradient gels were ordered from Gen-Script. As host strain for recombinant DNA and for protein expression, *Escherichia coli* NEB 10-beta (New England Biolabs) was used. Precultures were grown in lysogeny broth (LB), and the subsequent main cultures in terrific broth (TB) or ZYM505 medium (*Studier, 2005*).

## Cloning, expression and purification

The gene of ADHA was identified in a PCR-based genome screening for new ketoreductases from *Candida magnoliae* DSMZ 70638 using degenerate PCR primers. A gene fragment of a new ketoreductase was identified. By using a refined PCR-primer the full open reading frame of the previously unknown ketoreductase ADHA was identified. The gene encoding for ADHA was cloned into c-LEcta's proprietary expression vector pLE1A17 (a derivative of pRSF-1b, Novagen). After transformation of electrocompetent *E. coli* BL21 (DE3) cells with the pLE1A17-ADHA vector, the cells were cultivated in ZYM505 medium (*Studier, 2005*), supplemented with kanamycin (50 mg/L) at 37 ˚C. Expression was induced at logarithmic phase by IPTG (0.1 mM) and carried out at 30 ˚C for 16–18 hr. Cells were harvested by centrifugation (3220 x g, 20 min, 4 ˚C) and disrupted with cell lysis buffer (50 mM Tris-HCl pH 7.0; 2 mM $MgCl_2$, 1x CelLytic B (Sigma-Aldrich); DNA nuclease 0.02 U, lysozyme 0.5 mg/mL). The crude extracts were separated from cell debris by centrifugation (3220 x g 30 min, 4 ˚C). The supernatant was sterile-filtrated over a 0.2 μm-membrane filter and subsequently freeze dried. A brownish powder was obtained. This sample was purified by anion-exchange (Q Sepharose Fast Flow, GE Healthcare) and gel-permeation chromatography (Superdex 200, GE healthcare) in 50 mM Tris-HCl pH 7.5.

The genes encoding for the mutant enzymes were cloned into a vector with a cleavable tag (ADHA M9*: N-His-SUMO in pBAD vector). In this way, the proteins could be obtained in high purity, and, after cleavage of the tag, in the native form. A preculture inoculated with a single colony of *E. coli* NEB 10-beta harbouring the pBAD vector was grown overnight in LB (5 mL) with ampicillin (50 mg/L) at 37 ˚C. This preculture was used to inoculate TB (5 v/v %) and was simultaneously induced with 0.02% L-arabinose. The cells were grown at 24 ˚C for another 38 hr. Then the cells were spun down (3000 x g, 10 min, 4 ˚C), resuspended in buffer (50 mM Tris-HCl pH 7.5), and sonicated (5 s on/off, 70% amplitude) on ice for cell lysis. Debris was spun down (21000 x g, 45 min, 4 ˚C), and the supernatant was applied to an IMAC column (GE Healthcare). After washing with buffer, and buffer with 5 mM imidazole, the His-tagged protein was eluted with 300 mM imidazole in buffer. The sample was subsequently subjected to a gravity-flow desalting column (PD-10, GE healthcare), to remove the imidazole. In case of SUMO-tagged protein, the protein was then incubated overnight at 4 ˚C with SUMO protease. The next day, the cleaved protein was separated from the SUMO-protease by IMAC and further purified by gel-permeation chromatography (Superdex 200, GE healthcare) in 50 mM Tris-HCl pH 7.5. The protein samples were analysed by SDS-PAGE (*Supplementary file 2*).

## Crystallization and structure determination

Wild-type and mutant ADHA, purified as above, crystallized under several conditions. To probe crystallization conditions, several screens were preformed using an Oryx eight robot (Douglas Instruments, UK) starting from PEG Suite, Ammonium Sulphate Suite and Classic Suite (Nextal). Preliminary crystallization experiments were performed in sitting drop using MRC SWISS CI plates, with a 1:1 protein:reservoir ratio and 0.4–1 mL crystallization droplets. Crystallization experiments were performed also with proteins incubated with 1 mM NADP$^+$. The best crystals grew in 20% w/v PEG 3350, 0.2 M sodium citrate. These conditions were then used for all (wild type and mutant) crystallization experiments. Crystals were harvested from crystallization droplets using nylon cryo-loops (Hampton Research/Molecular Dimensions), shortly soaked in a cryo-protectant solution containing 20% w/v PEG 3350, 0.2 M sodium citrate and 20–25% v/v PEG 400, and then flash-cooled in liquid nitrogen. Data collection was performed at ESRF ID23 beam line (European Synchrotron Radiation Facility, Grenoble, France) for the wild-type crystals, and at the Swiss Light Synchrotron (Paul Scherrer Institut, Villigen, SUI) for the M9 mutant. Wild-type crystals diffracted at 1.6–2.0 Å, while the resolution of diffraction by the mutant crystals was 2.6 Å. Data processing, structure determination by molecular replacement, and structure refinement were performed with XDS and programs of the CCP4 suited using standard protocols (*Kabsch, 1993*; *Winn et al., 2011*). The crystallographic statistics are listed in *Supplementary file 1A*.

## Computational methods

Calculations for the FRESCO workflow to predict stabilizing mutations were carried out as described (*Wijma et al., 2018*). Although we also obtained a crystal structure of ADHA in complex with NADP$^+$, we chose the wild-type structure without the ligand for our calculations, because of two reasons: firstly, the FoldX and Rosetta software only accept protein residues and the cofactor would have to be omitted for these parts in any case. Secondly, NADP$^+$ is assumed to interact only transiently to the enzyme and does not bind permanently as a prosthetic group. Thus, to realistically model the enzyme in its most relevant physiological condition, we did not consider the ligand-bound structure for the MD simulations either. In brief, we then prepared the structure using YASARA by adding hydrogens, predicted their optimal H-bonding network, and removed buffer and water molecules from the structure. The Rosettaddg and FoldX algorithms were then applied as described, and the resulting list of energies was used to identify 478 mutations with a $\Delta\Delta G^{\text{Fold}}$ improvement of < −5 kJ mol$^{-1}$. We re-added the water molecules and furthermore solvated the protein in a charge-neutralized simulation cell stretching 7.5 Å around all atoms. MD simulations were carried out with YASARA using the Yamber3 force field. In five independently seeded simulations, the system was first energy minimized, heated to 298 K in 30 ps, equilibrated for 20 ps, and finally simulated in the production run for 50 ps. The MD trajectories' average structures were superimposed and compared with those of the wild type in a visual inspection step of all MD-simulated mutants. Taking into account both the predicted static, as well as the MD simulated structures, mutants were not

considered for experimental verification if we observed an increased backbone or side-chain flexibility, hydrogen bond impairment, or hydrophobic exposure. The Peregrine high-performance computing cluster (University of Groningen, Groningen) was used to perform all calculations for Rosettaddg, FoldX, and the molecular dynamics (MD) simulations. The MD simulations were carried out using the software Yasara Structure (YASARA Biosciences GmbH).

## Mutagenesis and small-scale growth and expression

Following the published protocol (*Fürst et al., 2018*), mutant oligonucleotides were designed and ordered, and the mutant library was prepared via QuikChange. QuikChange was also used to prepare mutants in which several mutations were combined. The construct used for small-scale expression of the mutants was an L-arabinose-inducible pBAD vector, where the ADHA gene is fused to an N-terminal His-tag. The plasmid was used to transform chemically competent *E. coli* NEB 10 beta cells. Cell cultures were grown in four 96-deepwell plates, containing 1 mL per well. When the 1 mL TB medium was inoculated with an overnight preculture (5% v/v), arabinose was also added, and the 96-deepwell plates were incubated for 38 h at 24 °C. Then, the cells were centrifuged to create a single pellet for each mutant from 4 mL of culture broth. Cells were resuspended in 50 mM Tris-HCl pH 7.5. After lysis as described (*Fürst et al., 2018*), His-tag purification and desalting, the samples were used for determining activity and thermostability melting point determination.

## Melting point determination by ThermoFluor

For probing the thermostability of each mutant, the apparent melting temperatures were measured by the ThermoFluor method (*Pantoliano et al., 2001*). For this, purified enzyme was mixed with the SYPRO orange dye and the apparent $T_m$ was measured in an RT-PCR (CFX96 Touch Real-Time PCR, BioRad, signal setting: FRET), in accordance with the described method (*Cummings et al., 2006*). All 177 enzyme mutants were screened in technical duplicate.

## Activity assay

Activity was measured by monitoring the formation of NADPH at 340 nm. After mixing enzyme ($\leq$0.1 µM) with substrate (cyclohexanol, if not stated: at 70 mM) in buffer (50 mM Tris-HCl pH 7.5), 100 µM $NADP^+$ was added, briefly mixed in a cuvette, and then the reaction was followed (V-330 Spectrophotometer, JASCO). The slopes of the initial 20 s were used to determine the rates, in $\Delta$Abs/min. This value was then divided by the extinction coefficient of NADPH ($\varepsilon_{340}$ = 6.22 mM$^{-1}$ cm$^{-1}$), in accordance with the Lambert-Beer law, resulting in a rate in mM/min. By dividing this value by the protein concentration in the reaction, the $k_{obs}$ values were obtained. All measurements were done in technical duplicates or triplicates.

## Temperature stability

Dilutions of wild-type and mutant ADHA were made in buffer (50 mM Tris-HCl pH 7.5), to a concentration of 0.15–0.3 mg/mL. These were incubated in a water bath at 37 °C. At several intervals, samples were taken, mixed with buffer and substrate (70 mM cyclohexanol) that were prewarmed at 37 °C in the same water bath. To start the reaction, cofactor (200 µM $NADP^+$) was added. Absorbance at 340 nm was monitored (V-330 Spectrophotometer, JASCO), and activity was determined based on the slope.

## Conversions and enantiomeric excess

A solution of ethyl-4-chloro-3-oxobutanoate (COBE) in DMSO (50 µL, 200 mM) was added to a Tris-HCl buffer (950 µL, 50 mM, pH 7.5) containing 0.5 mM NADPH, 5 µM of ADHA (WT or M9*), 5 µM of PTDH (phosphite dehydrogenase from *Pseudomonas stutzeri* WM88) and 50 mM sodium phosphite. The mixture was incubated for 24 h in an orbital shaker (150 rpm, 25 °C). The solution was extracted with ethyl acetate (EtOAc) (3 × 1 mL), centrifuging after each extraction (10000 x g, 2 min), and the combined organic solutions were dried over anhydrous $Mg_2SO_4$, and analyzed by chiral GC. Enantiomeric excess determinations for COBE were measured by chiral GC-FID. GC–FID analyses were carried out with an Agilent Technologies 7890A GC system. In *Supplementary file 4* chromatograms and details are provided regarding the settings for the GC.

A solution of 4-chloroacetophenone (4-CAP) in DMSO (50 µL, 200 mM) was added to a Tris-HCl buffer (950 µL, 50 mM, pH 7.5) containing 0.5 mM NADPH, 5 µM of ADH, 5 µM of PTDH and 50 mM sodium phosphite. The mixture was incubated for 24 h in an orbital shaker (150 rpm, 25 ℃). The solution was extracted with EtOAc (3 × 1 mL), centrifuging after each extraction (10000 x g, 2 min), and the combined organic solutions were dried over anhydrous $Mg_2SO_4$. Organic solvent is evaporated under reduced pressure and residue is dissolved in 3 mL heptane/iso-propanol (v/v = 95:5) and analyzed by chiral HPLC. Enantiomeric excess determinations for 4-CAP were measured by normal phase (np) HPLC analysis (Chiralcel OD-H column) using UV-detection (Shimadzu SCL-10Avp).In *Supplementary file 4* chromatograms and details are provided regarding the settings for the HPLC.

## Acknowledgements

This research received funding from the European Union (EU),project ROBOX (grant agreement no. 635734) under the EU's Horizon 2020 Program Research and Innovation actions H2020-LEIT BIO-2014–1 and from the Italian Ministry of Education, University and Research (MIUR), Dipartimenti di Eccellenza Program (2018–2022) – Department of Biology and Biotechnology 'L. Spallanzani', University of Pavia- The views and opinions expressed in this article are only those of the authors and do not necessarily reflect those of the European Union Research Agency. The European Union is not liable for any use that may be made of the information contained herein.

## Additional information

### Competing interests

Sebastian Bartsch: A patent application on the original ADH was filed by c-LEcta (WO 2019/012095). The other authors declare that no competing interests exist.

### Funding

| Funder | Grant reference number | Author |
| --- | --- | --- |
| European Commission | EU-H2020-ROBOX grant agreement nr. 635734 | Friso S Aalbers<br>Maximilian JLJ Fürst<br>Stefano Rovida<br>J Rubén Gómez Castellanos<br>Sebastian Bartsch<br>Andreas Vogel<br>Andrea Mattevi<br>Marco W Fraaije |

The funders had no role in study design, data collection and interpretation, or the decision to submit the work for publication.

### Author contributions

Friso S Aalbers, Conceptualization, Data curation, Formal analysis, Investigation, Methodology, Writing - original draft; Maximilian JLJ Fürst, Software, Visualization, Writing - review and editing; Stefano Rovida, Formal analysis, Investigation, Methodology; Milos Trajkovic, Investigation, Methodology; J Rubén Gómez Castellanos, Data curation, Formal analysis, Investigation; Sebastian Bartsch, Data curation, Investigation, Methodology, Writing - review and editing; Andreas Vogel, Methodology, Writing - review and editing; Andrea Mattevi, Conceptualization, Supervision, Writing - review and editing; Marco W Fraaije, Conceptualization, Supervision, Funding acquisition, Project administration, Writing - review and editing

### Author ORCIDs

Friso S Aalbers ![ORCID] https://orcid.org/0000-0002-2142-9661
Maximilian JLJ Fürst ![ORCID] https://orcid.org/0000-0001-7720-9214
Andrea Mattevi ![ORCID] http://orcid.org/0000-0002-9523-7128
Marco W Fraaije ![ORCID] https://orcid.org/0000-0001-6346-5014

Decision letter and Author response
Decision letter https://doi.org/10.7554/eLife.54639.sa1
Author response https://doi.org/10.7554/eLife.54639.sa2

## Additional files

### Supplementary files

• Supplementary file 1. Crystallographic data and activity-impairing mutations. (**A**) Crystallographic data. (**B**) Table of stabilizing mutations that disrupt activity.

• Supplementary file 2. SDS-PAGE analysis. (**A**) Expression of ADHA at different temperatures. (**B**) Purification of SUMO-M9*.

• Supplementary file 3. Correlation of predictions with the measured data. (**A**) Correlation of predicted values and experimental melting point data. (**B**) Graph of correlation.

• Supplementary file 4. Chromatograms of conversions. (**A**) Settings for GC and HPLC. (**B**) Chromatograms GC. (**C**) Chromatograms HPLC.

• Transparent reporting form

### Data availability

Diffraction data have been deposited in PDB under the accession codes 6TQ3, 6TQ5, and 6TQ8. Details on the structures, enzyme kinetic data and statistical analyses are included in the supplementary information.

The following datasets were generated:

| Author(s) | Year | Dataset title | Dataset URL | Database and Identifier |
| --- | --- | --- | --- | --- |
| Rovida S, Aalbers FS, Fraaije MW, Mattevi A | 2019 | Alcohol dehydrogenase from Candida magnoliae DSMZ 70638 (ADHA) | https://www.rcsb.org/structure/6TQ3 | RCSB Protein Data Bank, 6TQ3 |
| Rovida S, Aalbers FS, Fraaije MW, Mattevi A | 2019 | Alcohol dehydrogenase from Candida magnoliae DSMZ 70638 (ADHA): complex with NADP+ | https://www.rcsb.org/structure/6TQ5 | RCSB Protein Data Bank, 6TQ5 |
| Rovida S, Aalbers FS, Fraaije MW, Mattevi A | 2019 | Alcohol dehydrogenase from Candida magnoliae DSMZ 70638 (ADHA): thermostable 10fold mutant | https://www.rcsb.org/structure/6TQ8 | RCSB Protein Data Bank, 6TQ8 |

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
