## [Decision Letter]

**Acceptance summary:**

This study applies the FRESCO computational method to dramatically enhance the thermal stability of a dehydrogenase enzyme that has potential importance in the biotransformation of industrially relevant compounds. The use of X-ray crystal structural analysis of the optimized enzyme helps explain its exceptional properties. This study serves as an excellent model for what is possible in the thermal engineering of proteins using computational design.

**Decision letter after peer review:**

Thank you for submitting your article "Approaching boiling point stability of an alcohol dehydrogenase through computationally-guided enzyme engineering" for consideration by *eLife*. Your article has been reviewed by three peer reviewers, and the evaluation has been overseen by Reviewing/Senior Editor Philip Cole. The reviewers have opted to remain anonymous.

The reviewers have discussed the reviews with one another and the Reviewing Editor has drafted this decision to help you prepare a revised submission.

Summary:

This is an interesting paper that describes the successful application of FRESCO, a computational method for protein stabilization previously reported by the authors, for the stabilization of an alcohol dehydrogenase. Upon screening > 150 computationally selected mutations and combining the most stabilizing ones, the authors were able to obtain a 51 °C stabilization compared to the parent enzyme, which is an impressive achievement. By removing a stabilizing but deleterious mutation, the authors were able to achieve a significant level of stabilization without losing catalytic activity. This work encompasses a significant amount of work and the results are relevant and important. The structural analyses on the evolved variant nicely complement the characterization studies.

Essential revisions:

– The authors should clarify how accessible FRESCO is to the scientific community in terms of cost and technical aspects

– Have the authors determined the *ee* values for a range of substrates for their most thermostable enzyme? One worries that in the process of incorporating these mutations, stereochemical selectivities associated with the wt enzyme may be reduced or otherwise altered.

– All tables and graphs except Figure 5 are missing error bars. These should be included.

– "Since we suspected the S197E mutation to be problematic concerning coenzyme binding, based on the structural observations (Figure 3C),...". This is an important point that could be explained more clearly. It is not obvious from the Figure 3C why the S197E mutation should be deleterious for cofactor binding. Due to effect of mutation on the flexibility of the loop? What insights do the MD simulations provide into that? The E197 neg charge could cause electrostatic repulsion with phosphate of NADPH, but what is distance from E197 and the cofactor? And isn't E197 neg charge neutralized by interaction with Arg R18, as discussed in the manuscript? The electrostatic repulsion effect could be probed by comparing/contrasting activity of WT and mutant using NADH.

– "The initial FRESCO calculations and predictions for ADHA had a good hit rate: of the tested 177 mutations, 57 had a stabilizing effect of > 1 °C." This number should be adjusted considering error in Tm measurement (+/- 1C). Because of the error, it seems more appropriate to indicate # of mutants with DeltaTm> 2degC as significant.

– There is barely any reference to literature related to the use of computational design for enzyme stabilization. Below are some relevant papers that should be mentioned/cited (among others):

Malakauskas and Mayo, 1998; Bjørk et al., 2004; Korkegian et al., 2005; Gribenko et al., 2009; Borgo and Murphy, 2012; Bednar et al., 2015; Moore et al., 2017; Shah et al., 2007; Murphy et al., 2002.

Depending on the outcome of a thorough literature search, the authors might wish to revise the statement, "of all studies on stabilization of enzymes or proteins to date, to the best of our knowledge".

– Re: "The structure of ADHA bound to NADP^+^ unveils a well-defined catalytic pocket". This seems to be contradicted by the note in Figure 1 that the nicotinamide moiety is disordered.

– There is an unclear methodology aspect of the (probably too brief) introduction to FRESCO. At the outset it was not clear to what extent the coenzyme was included in the folding free energy or MD calculations. Near the end of the paper the authors say, "the employed computational algorithms (Rosettaddg and FoldX) cannot take into account the flavin-protein interactions (Wijma et al., 2018)." Completely omitting the coenzyme during ddG and MD calculations is a surprising enough action that the authors should be very explicit in the methods description.

– Another methodology clarification request. It is not clear whether visual inspection occurred for the 478 design models, the 478 MD simulations, or the 478 MD simulation final snapshots. The length of the MD simulations is either not mentioned or not sufficiently prominent.

– It is unclear if there was a quantitative threshold used to filter undesirable mutations? Similarly, when choosing 2 or 3 out of 4 beneficial mutations, how was the choice made? algorithm or biophysical intuition? Similarly, was the final roster of 177 mutations based on an unbiased, quantitative cutoff?

– Currently the comparison between panels A and B for "Figure 3. Michaelis-Menten" is awkward, since not only are there scale changes in both axes but also a unit change that might be overlooked.

– Re: "The interaction it would have with R18, when in the context of the other introduced mutations, was impossible predict". You mean to say, "impossible to predict", but might not more sampling reveal the new favored conformation? This seems like an excellent test case for developing improved FRESCO methods.

– The 6TQ8 difference between R-work and R-free is 10%, which seems rather large. We worry that there might be a significant level of bias in that model. The wwPDB validation report seemingly backs this up as a concern.

---

## [Author Response]

Essential revisions:– The authors should clarify how accessible FRESCO is to the scientific community in terms of cost and technical aspects.

We have added the following sentences in the section of the Results on mutant prediction:

“A detailed description of the procedure is available as a step-by-step protocol suitable for biochemists with minimal computational experience and all the scripts are deposited online (Wijma et al., 2018). […] The experimental part has also been described in great detail (Fürst et al., 2018) and requires readily purchasable consumables and equipment commonly available in most labs.”

– Have the authors determined the ee values for a range of substrates for their most thermostable enzyme? One worries that in the process of incorporating these mutations, stereochemical selectivities associated with the wt enzyme may be reduced or otherwise altered.

The *ee* values for two ketone reductions with the WT and mutant have been determined and added (Table 3). The enantioselectivity was tested for the reduction of aromatic and aliphatic compounds: in both cases the enantioselectivity was fully retained. The wild type and mutant ADH preferentially generated the same enantiomer of the corresponding alcohol with excellent *ee* values > 97%. For the raw data, an additional supplementary file has been created (Supplementary file 4).

– All tables and graphs except Figure 5 are missing error bars. These should be included.

Error bars have been added where they were missing.

– "Since we suspected the S197E mutation to be problematic concerning coenzyme binding, based on the structural observations (Figure 3C),...". This is an important point that could be explained more clearly. It is not obvious from the Figure 3C why the S197E mutation should be deleterious for cofactor binding. Due to effect of mutation on the flexibility of the loop? What insights do the MD simulations provide into that? The E197 neg charge could cause electrostatic repulsion with phosphate of NADPH, but what is distance from E197 and the cofactor? And isn't E197 neg charge neutralized by interaction with Arg R18, as discussed in the manuscript? The electrostatic repulsion effect could be probed by comparing/contrasting activity of WT and mutant using NADH.

The negative effect with respect to NADP binding arises from a combination of these effects: (1) dislocating the loop and positioning of E197 at the pyrophosphate binding site, thereby blocking the space for NADP^+^ to bind, and (2) interaction of E197 with R18, that should coordinate the 2’ phosphate of NADP^+^. The MD simulations were conducted without NADP, and the mutant structure also doesn’t contain the cofactor. Activity with NADH could not be probed, since neither WT or M9 are active with NADH.

We have changed the main text on various positions to clarify this point:

“The only considerable backbone shift (up to 3.1 Å) concerns loop 196-214 (Figure 3B, C). […] While the favourable electrostatics of this new interaction is a likely explanation for the mutation’s stability effect, it also explains the lower affinity for NADP^+^ (and the absence of the cofactor in the mutant’s crystals), as the salt bridge neutralizes R18 and occupies the NADP^+^ binding pocket.”

“Figure 3. Structure of the M9 mutant of ADHA, with mutated resides highlighted. **[…]** As a result, the cofactor (green carbons) is only bound in the wild type, while absent from the mutant structure.”

“Rescue of activity

Since we observed and suspect that the S197E mutation recruits R18 into the cofactor binding pocket and thereby impairs NADP^+^ binding (Figure 3C), we reverted this mutation.”

– "The initial FRESCO calculations and predictions for ADHA had a good hit rate: of the tested 177 mutations, 57 had a stabilizing effect of >1 °C." This number should be adjusted considering error in Tm measurement (+/- 1C). Because of the error, it seems more appropriate to indicate # of mutants with DeltaTm> 2degC as significant.

The text has been adjusted in accordance with the suggestion: “The initial FRESCO calculations and predictions for ADHA had a good hit rate: of the tested 177 mutations, 39 had a stabilizing effect of >2 °C (22%). This is in line with previous uses of FRESCO, in which between 20-40% of the FRESCO-predicted mutations were found to be stabilizing (Fürst et al., 2018; Wijma et al., 2018).”

– There is barely any reference to literature related to the use of computational design for enzyme stabilization. Below are some relevant papers that should be mentioned/cited (among others):Malakauskas and Mayo, 1998; Bjørk et al., 2004; Korkegian et al., 2005; Gribenko et al., 2009; Borgo and Murphy, 2012; Bednar et al., 2015; Moore et al., 2017; Shah et al., 2007; Murphy et al., 2002.

The references were added to a new paragraph in the Introduction.

“Several studies over the past two decades have developed strategies to stabilize proteins and enzymes. […] Considering the large improvements in stability that were found using these methods, with minimal screening work in the lab, such computational approaches are highly appealing.”

Depending on the outcome of a thorough literature search, the authors might wish to revise the statement, "of all studies on stabilization of enzymes or proteins to date, to the best of our knowledge"

The text has been adjusted in accordance with the suggestion: “With the aim of creating a robust ADHA variant, we applied FRESCO and reached a 9-fold mutant that shows a significantly improved thermostability with largely retained activity (**Table 3**). This study achieved the highest improvement of melting temperature of all FRESCO studies to date.”

– Re: "The structure of ADHA bound to NADP^+^ unveils a well-defined catalytic pocket". This seems to be contradicted by the note in Figure 1 that the nicotinamide moiety is disordered.

Considering that the majority of the NADP^+^ structure is confidently placed, the pocket of residues surrounding the disordered nicotinamide moiety, including the catalytic triad, is well-defined. For clarity, the text was changed: “The structure of ADHA bound to NADP^+^ unveils, apart from a disordered nicotinamide moiety, a well-defined catalytic pocket with a size suited for small cyclic compounds such as cyclohexanol, a good substrate of the enzyme.”

– There is an unclear methodology aspect of the (probably too brief) introduction to FRESCO. At the outset it was not clear to what extent the coenzyme was included in the folding free energy or MD calculations. Near the end of the paper the authors say, "the employed computational algorithms (Rosettaddg and FoldX) cannot take into account the flavin-protein interactions (Wijma et al., 2018). " Completely omitting the coenzyme during ddG and MD calculations is a surprising enough action that the authors should be very explicit in the methods description.

We agree with the reviewers that this should have been more explicitly mentioned and have revised the Materials and methods section to explain our reasoning behind using the ligand-free structure for the calculations: “Calculations for the FRESCO workflow to predict stabilizing mutations were carried out as described (Wijma et al., 2018). […] The MD simulations were carried out using the software Yasara Structure (YASARA Biosciences GmbH).”

We also added the information to the Results section:

“FRESCO computational predictions

Because the energy prediction algorithms rosetta and FoldX do not accept non-proteinogenic residues and because the ligand-free enzyme represents the physiologically more relevant form, we used the crystal structure without NADP for the stabilizing mutation predictions. While FRESCO usually excludes residues of ligand-binding sites, we decided to include such residues for predicting stabilizing mutations.”

– Another methodology clarification request. It is not clear whether visual inspection occurred for the 478 design models, the 478 MD simulations, or the 478 MD simulation final snapshots. The length of the MD simulations is either not mentioned or not sufficiently prominent.

Please see our answer and the revised text in the previous comment to see how we addressed this issue in the revised version.

– It is unclear if there was a quantitative threshold used to filter undesirable mutations? Similarly, when choosing 2 or 3 out of 4 beneficial mutations, how was the choice made? algorithm or biophysical intuition? Similarly, was the final roster of 177 mutations based on an unbiased, quantitative cutoff?

Again, we believe that our revised Materials and methods section answers this question, see above. As described in more detail in Wijma et al., 2018 (which we refer to various times in the manuscript), the inspection step is performed by the researcher and thus arguably subjective, although of course based on biophysical considerations.

When choosing representatives from multiple mutation suggestions for the same residue, we chose to test only one from the commonly considered subgroups of amino acids (hydrophobic, negatively/positively charged, etc.). This approach, together with outlined biophysical considerations yielded the selection of 177 mutants.

– Currently the comparison between panels A and B for "Figure 3. Michaelis-Menten" is awkward, since not only are there scale changes in both axes but also a unit change that might be overlooked.

The figures have been changed to have identical Y-axis (both units and scaling). For clarity, an inset has been included in the mutant Michaelis-Menten kinetics plot to illustrate the saturation behavior of the curve.

– Re: "The interaction it would have with R18, when in the context of the other introduced mutations, was impossible predict". You mean to say, "impossible to predict", but might not more sampling reveal the new favored conformation? This seems like an excellent test case for developing improved FRESCO methods.

Thank you pointing out the typo, which has been corrected. Indeed, it would hypothetically be possible to predict such a newly adopted conformation if either the prediction algorithms would be more accurate and allow backbone shifts, or if the MD simulation could sample a longer timeframe. With the current method of very short MD simulations (the price we pay for being able to simulate a large number of mutants), such large shifts are in fact, however, quite impossible to predict. As computational power increases, future applications of FRESCO may consider increased sampling time, yet, we are sceptical that such particular predictions are feasible (at least for academic research) within such a multi-mutant simulation framework in the near future.

– The 6TQ8 difference between R-work and R-free is 10%, which seems rather large. We worry that there might be a significant level of bias in that model. The wwPDB validation report seemingly backs this up as a concern.

The crystals of the thermostable mutant (6TQ8) contain four enzyme chains in the asymmetric unit. Therefore, we could apply density averaging and NCS restrains during structure determination and refinement. We feel that the structure is well refined and model bias is not an issue, particularly in light of the four-fold averaging. The somewhat high free Rfactor most likely reflects some disorder, especially in subunit B. We are submitting the PDB and structure factor files for inspection by the reviewer if she/he would like to directly look at the data.